# Simultaneous Determination of Dopamine and Uric Acid in Real Samples Using a Voltammetric Nanosensor Based on Co-MOF, Graphene Oxide, and 1-Methyl-3-butylimidazolium Bromide

**DOI:** 10.3390/mi13111834

**Published:** 2022-10-27

**Authors:** Maryam Roostaee, Hadi Beitollahi, Iran Sheikhshoaie

**Affiliations:** 1Department of Chemistry, Faculty of Science, Shahid Bahonar University of Kerman, Kerman 76175-133, Iran; 2Environment Department, Institute of Science and High Technology and Environmental Sciences, Graduate University of Advanced Technology, Kerman 7631885356, Iran

**Keywords:** dopamine, uric acid, graphene oxide, metal-organic framework, modified electrode, carbon paste electrode

## Abstract

A chemically modified carbon paste electrode, based on a CoMOF-graphene oxide (GO) and an ionic liquid of 1-methyl-3-butylimidazolium bromide (CoMOF-GO/1-M,3-BB/CPE), was fabricated for the simultaneous determination of dopamine (DA) and uric acid (UA). The prepared CoMOF/GO nanocomposite was characterized by field emission-scanning electron microscopy (FE-SEM), the X-ray diffraction (XRD) method, a N_2_ adsorption–desorption isotherm, and an energy dispersive spectrometer (EDS). The electrochemical sensor clearly illustrated catalytic activity towards the redox reaction of dopamine (DA), which can be authenticated by comparing the increased oxidation peak current with the bare carbon paste electrode. The CoMOF-GO/1-M,3-BB/CPE exhibits a wide linear response for DA in the concentration range of 0.1 to 300.0 µM, with a detection limit of 0.04 µM. The oxidation peaks’ potential for DA and uric acid (UA) were separated well in the mixture containing the two compounds. This study demonstrated a simple and effective method for detecting DA and UA in real samples.

## 1. Introduction

Dopamine, a key catecholamine neurotransmitter, can be found in a variety of body fluids, including urine, blood serums, and other bodily fluids [1,2]. It is involved in motor function, knowledge, memory, cognition, motivation, and learning, as well as regulating human behaviors and emotions [3,4,5]. Furthermore, the dopamine (DA) hydrochloride salt used in cardioversion shock treatment has been associated with heart attacks, kidney failure, and bacterial infections [6]. An unnatural DA concentration is a sign of a neurological disorder such as HIV infection, schizophrenia, Alzheimer’s, kidney disease, Parkinson’s disease, among other diseases [7,8,9]. Due to the low concentration of DA in the brain, accurate measurements of DA in biological samples are required for the diagnosis of these diseases [10].

However, in biological fluids, DA frequently coexists with high concentrations of uric acid [11]. Uric acid (UA) is a heterocyclic compound formed by the metabolism of purine nucleosides, guanosine, and adenosine [12]. Abnormal UA concentrations may be indicative of the symptoms of several diseases, including hyperuricemia, gout, and Lesh–Nyhan syndrome [13]. As a result, the sensitive and simultaneous detection of DA and UA is essential in a variety of fields, including biomedical chemistry, neurochemistry, and disease diagnosis investigations.

There are many techniques that are currently used for the simultaneous determination of DA and UA detection, such as mass spectrometry [14], spectrophotometry [15], fluorescence [16], capillary electrophoresis [17], high performance liquid chromatography (HPLC) [18], and so on. Although these procedures have been used for many years, they do have certain disadvantages, such as being expensive, time-consuming, and difficult to conduct. They also require a huge number of solvents and they are poisonous [12,19]. Among these techniques, electrochemical detection has received widespread attention for the detection of DA and UA due to its simplicity of use, low cost, and high sensitivity [20]. On the other hand, the overlapping of the oxidation peaks of DA and UA is a problem that needs solving in the selective or simultaneous detection of DA and UA [21]. As a result, it is critical to develop an effective nanomaterial for electrode modification in order to effectively separate the oxidation peaks and obtain simultaneous detection [22,23,24].

Recently, porous crystalline materials, composed of both inorganic and organic materials, such as metal–organic frameworks (MOFs), have been widely developed [25,26,27]. Topological diversity, high specific surface areas, aesthetically attractive structures, high porosity, and the great chemical and thermal stability of MOFs make them interesting to study [28,29]. Due to their virtuous properties, they have been broadly utilized in various areas such as drug delivery [30], energy storage [31], and gas storage [32], in addition to being used in colorimetric sensors [33], semiconductors [34], electrochemical sensors [35], and so on. On the other hand, the metal content of inorganic nodes in MOFs with active sites is a key factor in determining their catalytic performance [36,37]. Among these, Cobalt-based MOFs are of interest due to their unique properties, including high thermal stability, high specific surface areas, variable valency, and high porosity [28]; however, CoMOFs have an irregular crystalline shape, low conductivity, and are unstable in water, which restricts their potential application for use as an electrochemical sensor [38]. Therefore, many efforts have been made to modify CoMOFs, so that they can be used as high-performance electrochemical sensors. In this work, we used graphene oxide to increase the conductivity of the CoMOF.

Graphene oxide (GO) is a significant common material that has been researched and used in the electric field due to its excellent electrical conductivity and modifiability [39,40,41]. Through covalent bonding or coordination, surface functionalized graphene could be constructively coupled with some other molecules to create hybrid composites [42,43,44,45]. MOFs can indeed be uniformly distributed on graphene to create MOF/graphene oxide hybrid composites. MOF–GO hybrid substances improve performance by taking the benefits of the derived synergistic effect of the two constituent components [46].

Ionic liquids are organic compounds that are composed of organic cations with various anions, which are mostly liquid at room temperature. Due to their unique properties, including high thermal stability, relatively high ionic conductivity, less vapor pressure, good stability, biocompatibility, and good solvating properties, they have been utilized in the development of electrochemical sensors [47,48].

In this work, we have developed a CoMOF-GO and 1-methyl-3-butylimidazolium bromide with a modified CPE for the detection of dopamine and uric acid by exploiting the addictive properties of these two modifiers. This modified electrode is a high sensitivity electrochemical sensor that can be used for the simultaneous determination of DA and UA, and it showed two separated oxidation signals corresponding to these materials in the analyzed samples. The method was then examined for the voltametric determination of DA and UA in some real samples.

## 2. Experimental

### 2.1. Materials and Chemicals

Dopamine, uric acid, 1-methyl-3-butylimidazolium bromide, Co(NO_3_)_2_·6H_2_O, paraffin, and 1,3,5-benzenetricarboxylate (BTC) were purchased from Sigma Aldrich (St. Louis, MO, USA). Graphite powder, ethanol, *N*,*N*-dimethylformamide (DMF, >99.8), and methanol were obtained from Merck (Merck Chemicals GmbH, Darmstadt, Germany). The phosphoric acid (Merck) and sodium hydroxide (Sigma-Aldrich) were used in the preparation of the 0.1 M phosphate buffer solution (PBS) as the supporting electrolytes in the electrochemical determination of DA and UA. The water used in all experiments was deionized.

### 2.2. Apparatus

A SAMA 500 voltametric analyzer (Isfahan, Iran) was used for all electrochemical experiments which included cyclic voltammetry (CV), square wave voltammetry (SWV), and chronoamperometry. The electrochemical measurements were performed in an electrochemical cell (Metrohm AG, Herisau, Switzerland) equipped with a CoMOF-GO/1-M,3-BB/CPE working electrode, a platinum wire counter electrode (Azar Electrode, Urmia, Iran), and an Ag/AgCl/KClsat reference electrode electrode (Azar Electrode, Urmia, Iran). A pH meter (Metrohm, Model 827, Metrohm AG, Herisau, Switzerland) was used to measure the pH of the solutions.

The crystalline structures of the CoMOF and CoMOF-GO were examined by X-ray diffraction analysis using Cu-Kα radiation (X’PertPro (Etten Leur, The Netherlands)). The surface morphology and elemental analysis were examined using the FE-SEM Tescan Mira 3 (Bruker, Mannheim, Germany). N_2_ adsorption/desorption isotherms were measured, in temperatures equal to that of liquid nitrogen, using a Brunauer–Emmett–Teller device (BET, BELSORP MINI II, Japan). The BET method was used to calculate the specific surface areas of the samples.

### 2.3. Synthesis of the CoMOF

The CoMOF was synthesized using a similar procedure that was reported for having a nickel-based MOF [49]. Under sonication, 10.22 mmol Co(NO_3_)_2_.6H_2_O and 5.9 mmol BTC were dissolved in a volume ratio of 1:1:1 of ethanol, DMF, and deionized water. A uniform solution was transferred to an autoclave and heated to 120 °C in an oven for 125 h. Next, the sample was taken out of the oven and left to cool at room temperature. A violet precipitate was obtained and then rinsed several times with 25 mL DMF and 25 mL methanol. Finally, the prepared samples, as they were, were dried in a vacuum at 80 °C.

### 2.4. Synthesis of GO and CoMOF-GO

Graphene oxide was synthesized from graphite powder using a modified Hummers method [50]. In 25 mL of water, the CoMOF and GO were mixed in a 1:2 ratio to make the CoMOF-GO nanocomposite. The solution was sonicated for 3 h to achieve a uniform mixture. The prepared solution was then placed in an autoclave and kept at 100 °C for 24 h. The precipitate was centrifuged at 10,000 rpm, and the sample was washed at room temperature with deionized water and ethanol. Next, the sample had been dried in an oven at 60 °C.

### 2.5. Preparation of Modified Electrode

CoMOF-GO/1-M,3-BB/CPE was created by combining 0.05 g CoMOF-GO with 0.95 g of graphite powder, a suitable amount of paraffin oil as a binder, and 1-M,3-BB ionic liquid. These ingredients were thoroughly combined using a mortar and pestle for 40 min until an evenly wetted paste was produced. A copper wire was introduced into the paste as an electrical connection device, and the CoMOF-GO/1-M,3-BB paste was applied to the inside of a glass tube. A surplus of the paste was pushed out of the tube and polished on the weighing paper when a new surface was required.

### 2.6. Preparation of the Real Samples

Urine samples were obtained from a healthy volunteer. Urine samples were stored in a refrigerator immediately after collection. Ten milliliters of the sample were centrifuged for 15 min at 2000 rpm. The supernatant was filtered out using a 0.45 μm filter. Then, varying volumes of the solution were transferred into 25 mL volumetric flasks and diluted to the mark with PBS (pH 7.0). The diluted urine sample was spiked with different amounts of DA and UA. The DA and UA contents were analyzed using the proposed method and the standard addition method.

One milliliter of DA ampoule (Caspian Tamin Company, Tehran, Iran, contained 200 mg in 5 mL of dopamine) was diluted to 10 mL with 0.1 M PBS (pH 7.0); then, varying volumes of the diluted solution were transferred into a series of 25 mL volumetric flasks and diluted to the mark with PBS. The analysis of DA and UA was performed using the standard addition method.

## 3. Results and Discussion

### 3.1. Characterization of the CoMOF-GO Nanocomposite

The X-ray diffraction (XRD) pattern of the prepared materials is presented in Figure 1. XRD of the prepared nanocomposite was examined at 2θ, and it ranged from 5–80°. The X-ray diffraction of the prepared CoMOF is shown in Figure 1a. For the CoMOF, the peaks were located at 2θ = 8.12, 10.7, 17.02, 18.19, 24.5, and 28.6°. The XRD patterns of the CoMOF were quite comparable to those described by Cho et al. [51]. The XRD pattern of GO is presented in Figure 1b. The results show the diffraction peak at ~11.8° due to the (002) plane of GO [52]. Figure 1c also shows all the peaks related to the CoMOF and GO, with some baseline changes in comparison with those of the CoMOF, which show the successful synthesis of the CoMOF-GO nanocomposite.

N_2_ adsorption–desorption isotherms (Figure 2a) and Barrett–Joyner–Halenda (BJH) pore size distribution measurements (Figure 2b) were used to determine the specific surface area and porosity of the prepared CoMOF. The BET isotherm characteristic curve suggests a type III graph for the CoMOF. The key parameters of pore volume, and pore size for the sample were 0.05911 cm^3^·g^−1^ and 21.88 nm, respectively. The Lungmuir surface area of the CoMOF was measured at 343.29 m^2^·g^−1^. The BJH pore size distribution curve in Figure 2b further supports the sample’s mesoporous nature, which is mostly due to voids in the CoMOF. According to the BET data, the CoMOF has a large surface area and a perfect pore size, which would enhance electrochemical capacitance by increasing the number of ions available at the electrode/electrolyte interface.

Field emission-scanning electron microscopy (FE-SEM) is used to examine structural integrity and morphology. Figure 3a,b describes the FE-SEM morphology of the CoMOF. These images showed rods, rectangles, and squares that resemble 3D crystals, having a suitable length, width, and height, to confirm the CoMOF’s 3D structure. The EDS spectrum (Figure 4a) and elemental mapping images (Figure 4b) show that the Co, C, and O elements are distributed uniformly in the CoMOF product.

Figure 5a,b shows the FE-SEM images of the CoMOF/GO composite having the same three-dimensional structure as the CoMOF on the graphene oxide sheet. The embedded Co-MOF within the layers of the GO is seen in the images.

### 3.2. Electrochemical Behaviors of DA on the CoMOF-GO/1-M,3-BB/CPE

The electrochemical response of DA oxidation in the 0.1 M PBS, adjusted to variable pH values (3.0 to 10.0), was explored to determine the influence of the electrolyte solution’s pH. The results showed that the peak current of DA oxidation depended on the pH value; it reached a maximum as the pH increased up to 7.0, which then decreased as the pH value continued rising. Hence, the pH value of 7.0 was considered to be the optimum for subsequent electrochemical determinations.

The responses of the cyclic voltammogram for the oxidation of 150.0 µM DA at bare CPE (**curve a**), CoMOF/CPE (**curve b**), 1-M,3-BB/CPE (**curve c**), GO/CPE (**curve d**), CoMOF-GO/CPE (**curve e**), and CoMOF-GO/1-M,3-BB/CPE (**curve f**) are shown in Figure 6.

The results are shown in Table 1. According to the obtained result, CoMOF-GO/1-M,3-BB/CPE demonstrated the highest electrochemical activity for DA redox reaction because of the high porosity and abundance of active metal sites in MOF-derived structures.

### 3.3. Effect of Scan Rate on the redox reaction of Dopamine

CV was performed at various scan rates (20–400 mV·s^−1^) in 0.1 M PBS with 250.0 µM of DA to study the effect of scan rate on DA oxidation and the voltametric response of the CoMOF-GO/1-M,3-BB/CPE (Figure 7). The intensity of the peak and background current increases, with a small increase in peak potential when the scan rate is increased from 20 mVs^−1^ to 400 mVs^−1^. To investigate the relationship between scan rate and anodic and cathodic peak currents (Ipa, Ipc), a graph was drawn to highlight anodic and cathodic peak currents and the square root of the scan rate (*v*^1/2^) (Figure 7, inset). The findings revealed that the peak’s current increased linearly as the square root of the scan rate increased, which ranged from 20 to 400 mV·s^−1^, and the sensor’s operation had been controlled by the diffusion step for DA on the surface of CoMOF-GO/1-M,3-BB/CPE.

Figure 8 showed the Tafel plot for oxidation of 250.0 μM of DA at a scan rate of 20 mV/s. We calculated the electron transfer coefficient (α) using the slope of the Tafel plot, which equals n(1-α)F/2.303RT. With n = 1, the value of α was determined to be 0.43.

### 3.4. Chronoamperometric Analyses

Chronoamperometry was used to explore the DA catalytic oxidation on the CoMOF-GO/1-M,3-BB/CPE surface. Chronoamperometric analysis was conducted for variable DA content on the CoMOF-GO/1-M,3-BB/CPE using the working electrode potential of 170 mV. The chronoamperograms captured for the variable DA content on the CoMOF-GO/1-M,3-BB/CPE is seen in Figure 9. Cottrell’s equation (I = nFACD^1/2^ П^–1/2^t^−1/2^) explains the current (I) of an electrochemical reaction, of an electroactive material with a D value (diffusion coefficient), under a mass transport limited condition. Figure 9a shows a linear relationship between the I value with t^−1/2^ and the oxidation of variable DA content. The resulting slopes that correspond with the straight lines were plotted against the variable DA content (Figure 9B). The plotted slope and Cottrell’s equation estimated the D value to be 2.2 × 10^−6^ cm^2^/s for the DA.

### 3.5. Dynamic Range and Limit of Detection

SWV was employed to determine the DA in this study because it is highly sensitive to currents and it has a better resolution than cyclic voltammetry. The plot of I_pa_ vs. DA concentration showed a linear relation in the range of 0.1 µM–300.0 μM, and a detection limit of 0.04 µM was found for DA. In Table 2, the analytical characteristics of CoMOF-GO/1-M,3-BB/CPE was compared with prior published electrochemical sensors for DA detection, and the results show that CoMOF-GO/1-M,3-BB/CPE can perform very well compared with previous sensors in the determination of DA.

### 3.6. SWV Analysis for the Co-Detection of DA with UA

To confirm that CoMOF-GO/1-M,3-BB/CPE has the ability to co-detect DA with UA, the electrochemical responses of these analytes were detected by simultaneously changing the concentration of both analytes in PBS at pH 7.0. As seen in Figure 10, with concurrent changes in their concentrations, two non-interference peaks were found on the SWV curves. The peak currents for both DA and UA oxidation displayed a linear elevation for their respective concentrations (DA concentration range between 5.0 μM and 300.0 µM, and UA concentration range between 10.0 μM and 600.0 μM) (Figure 10A,B). The intensity of the peaks’ currents showed good linearity with the target concentration change, meaning that there is a possibility of detecting DA and UA in the blended solution.

### 3.7. Stability and Repeatability of CoMOF-GO/1-M,3-BB/CPE

SWV measurements of 50.0 µM DA were used to investigate the repeatability and stability of CoMOF-GO/1-M,3-BB/CPE. For ten replicate measurements, the relative standard deviation (RSD) was 3.7%, thus demonstrating significant detection repeatability. The CoMOF-GO/1-M,3-BB/CPE retains 95% of its initial response after 15 days and 90% after 30 days when stored in the laboratory. These findings indicate that the proposed electrode has excellent repeatability and stability.

### 3.8. Analysis of Real Specimens

The practical applicability of CoMOF-GO/1-M,3-BB/CPE was tested by sensing DA and UA in DA injections and urine specimens using the SWV procedure and standard addition method; this data can be seen in Table 3. The recovery rate is between 97.1% and 104.0%, and all RSD values were ≤3.5%. According to the experimental results, the CoMOF-GO/1-M,3-BB/CPE sensor possessed high potential for practical applicability.

Results of the urine analysis are shown in Figure 11.

## 4. Conclusions

In this work, the CoMOF/GO nanocomposite was prepared and then used to study the electrochemical behaviors of DA by CV, SWV, and chronoamperometry. We found that there was a high surface area and a synergistic impact between graphene oxide nanosheets and CoMOF, which provide a high sensing property and a fast electron transfer rate. The square wave voltametric response for DA is linear and in the dynamic range of 0.1–300.0 μM. The limit of detection is 0.04 μM. Moreover, the modified electrode exhibits effective electrocatalytic activity for the simultaneous determination of DA and UA. The proposed method could be used to determine DA and UA in real samples with promising results, and good recovery data were obtained.

## Figures and Tables

**Figure 1 micromachines-13-01834-f001:**
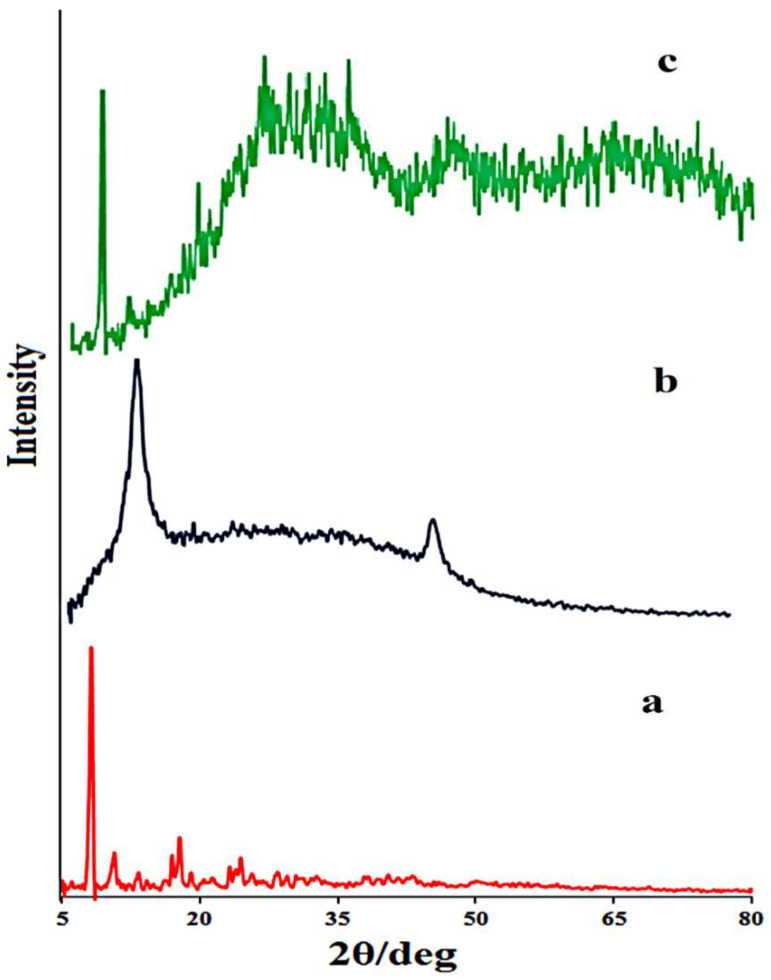
XRD patterns of the (**a**) CoMOF, (**b**) GO, and (**c**) CoMOF/GO nanocomposite.

**Figure 2 micromachines-13-01834-f002:**
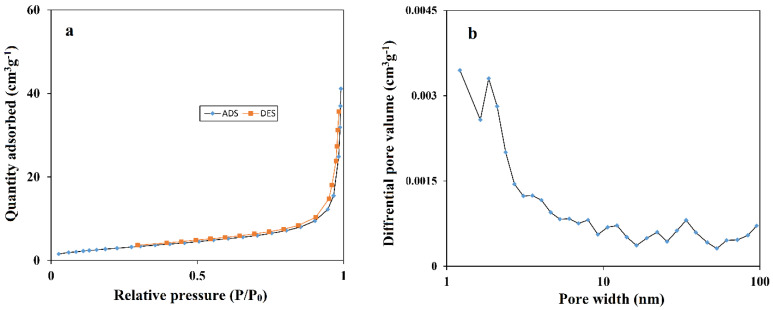
(**a**) N_2_ adsorption–desorption isotherms; and (**b**) BJH pore size distribution curve of the CoMOF.

**Figure 3 micromachines-13-01834-f003:**
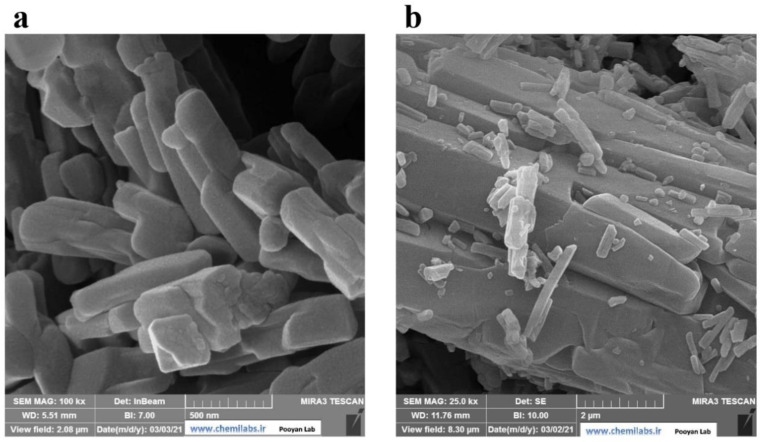
FE-SEM images of the CoMOF in (**a**) 500 nm and (**b**) 2 µM.

**Figure 4 micromachines-13-01834-f004:**
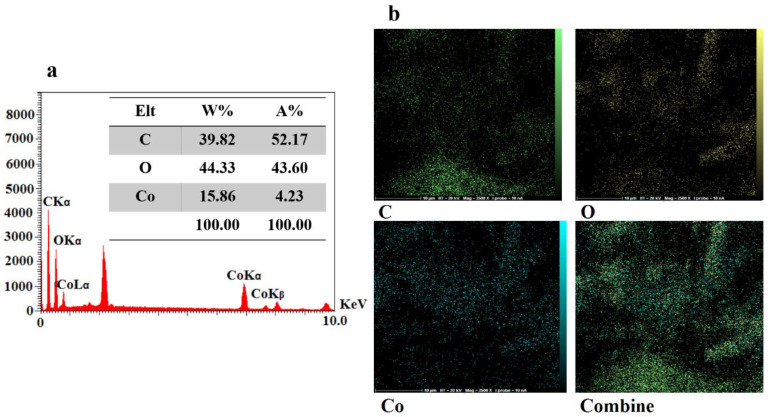
(**a**) EDS spectrum and (**b**) the corresponding elemental mapping images of Co, C, and O elements in the CoMOF sample.

**Figure 5 micromachines-13-01834-f005:**
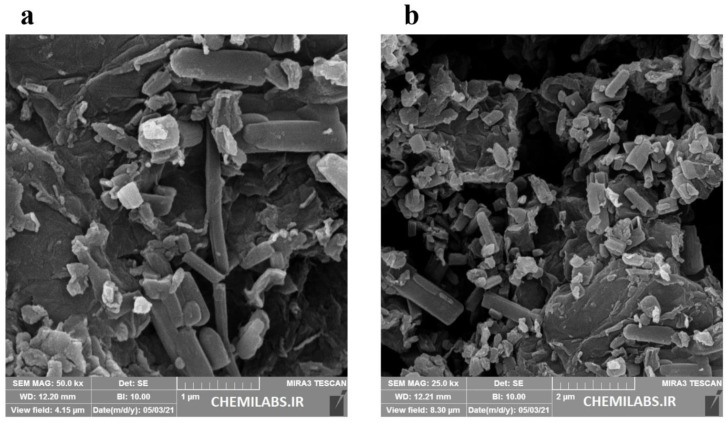
FE-SEM images of CoMOF-GO (**a**) 1 µM and (**b**) 2 µM.

**Figure 6 micromachines-13-01834-f006:**
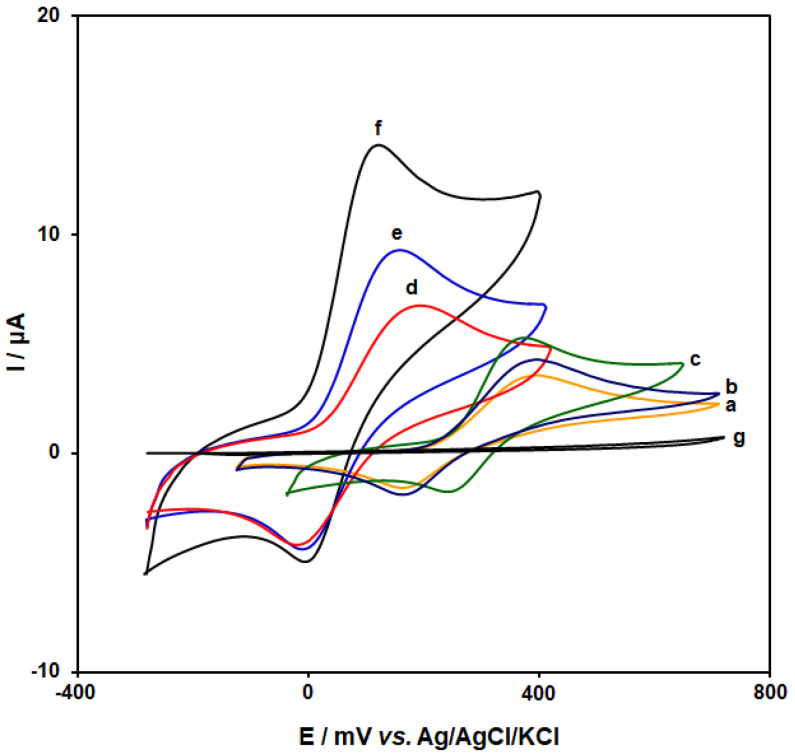
Cyclic voltammograms captured for the redox reaction of DA (150.0 μM) in PBS (0.1 M; pH = 7.0) on bare CPE (**curve a**), CoMOF/CPE (**curve b**), 1-M,3-BB/CPE (**curve c**), GO/CPE (**curve d**), CoMOF-GO/CPE (**curve e**), and CoMOF-GO/1-M,3-BB/CPE (**curve f**). Moreover, **curve g** shows the CoMOF-GO/1-M,3-BB/CPE in PBS (0.1 M; pH = 7.0). In all cases, the scan rate was 50 mV/s.

**Figure 7 micromachines-13-01834-f007:**
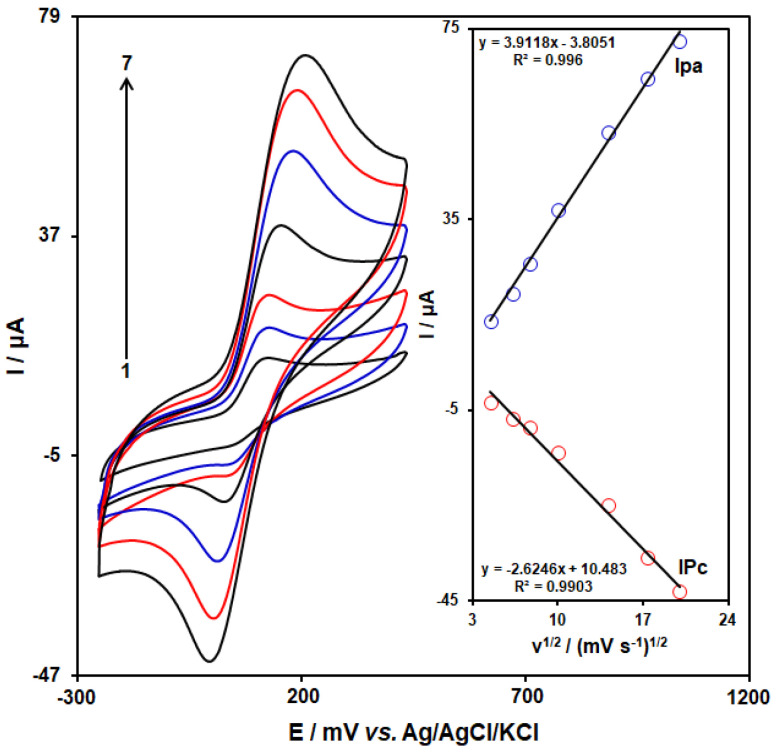
Cyclic voltammograms of CoMOF-GO/1-M,3-BB/CPE in 0.1 M PBS (pH 7.0) containing 250.0 µM DA at various scan rates; **1–7** correspond to 20, 40, 60, 100, 200, 300, and 400 mV·s^–1^, respectively. Inset: variation of anodic and cathodic peak currents vs. *v*^1/2^ for electrooxidation of 250.0 µM DA.

**Figure 8 micromachines-13-01834-f008:**
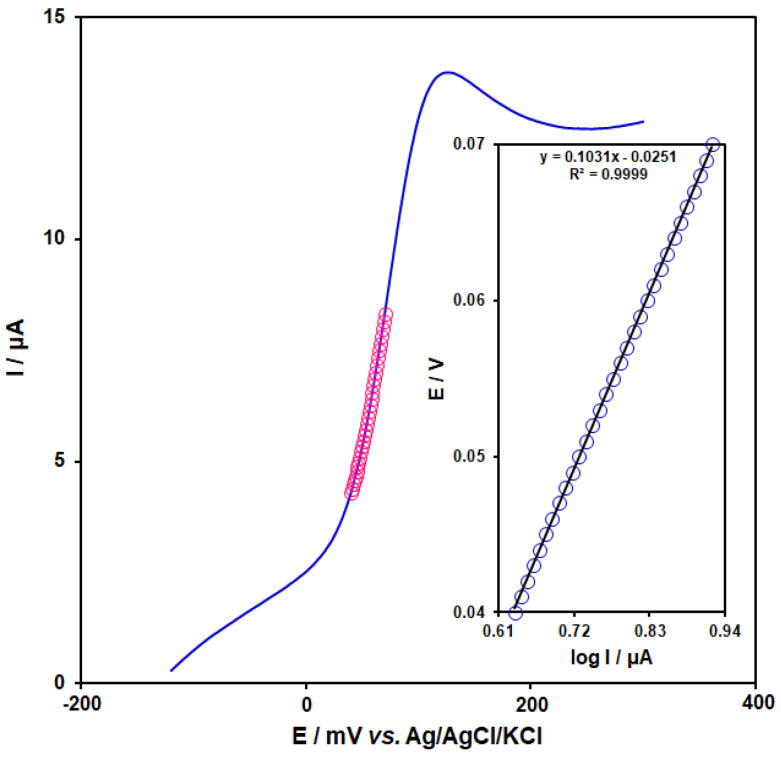
Linear sweep voltammogram response for CoMOF-GO/1-M,3-BB/CPE in 0.1 M PBS (pH 7.0) with a scan rate of 20 mV·s^−1^ in the presence of 250.0 μM DA (Circles are the Tafel region). Inset: the Tafel plot derived from the rising part or the corresponding voltammogram.

**Figure 9 micromachines-13-01834-f009:**
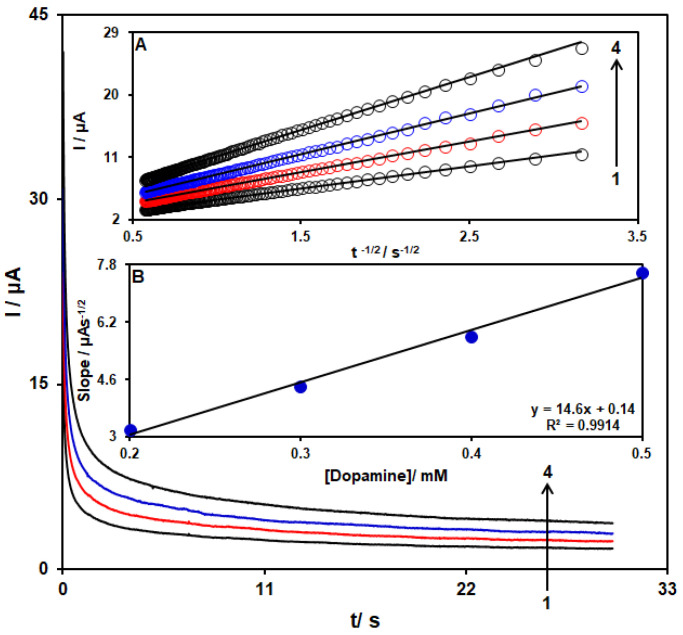
Chronoamperometric behavior of CoMOF-GO/1-M,3-BB/CPE in PBS (0.1 M; pH = 7.0) at potential of 170 mV for variable DA content. Insets: (**A**) plots of I vs. t^−1/2^ and (**B**) plots of the slopes from the straight lines vs. DA level (**1–4**: 0.2, 0.3, 0.4, and 0.5 mM).

**Figure 10 micromachines-13-01834-f010:**
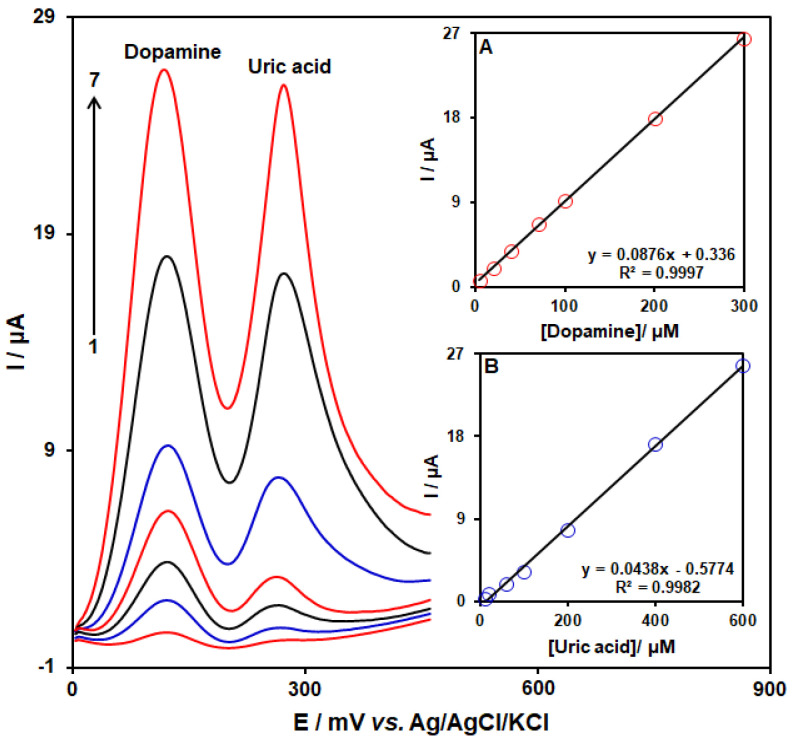
SWVs of CoMOF-GO/1-M,3-BB/CPE in PBS (0.1 M, pH = 7.0) with variable DA concentrations (**1–7**: 5.0, 20.0, 40.0, 70.0, 100.0, 200.0, and 300.0 μM) and UA (**1–7**: 10.0, 20.0, 60.0, 100.0, 200.0, 400.0, and 600.0 μM). Insets: (**A**) plot of peak current versus DA concentration, (**B**) plot of peak current versus UA concentration.

**Figure 11 micromachines-13-01834-f011:**
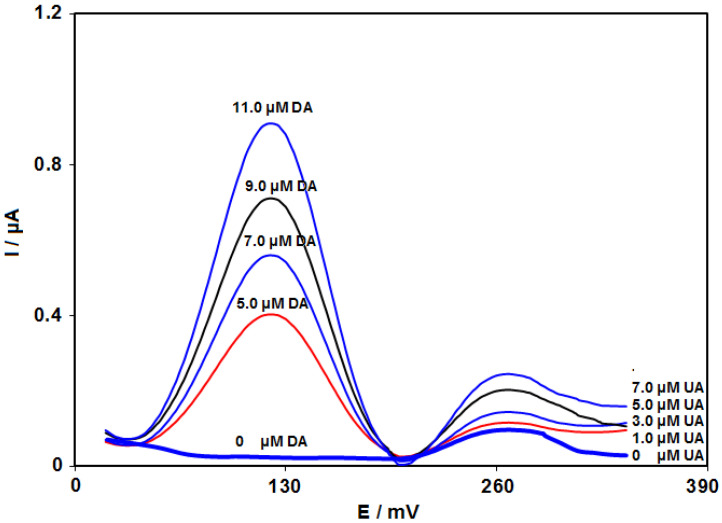
SWVs of CoMOF-GO/1-M,3-BB/CPE in urine samples with variable DA and UA concentrations.

**Table 1 micromachines-13-01834-t001:** Comparison of the electrochemical oxidation of DA (150.0 μM) in PBS (0.1 M; pH = 7.0) on the surface of various electrodes.

Electrode	Anodic Peak Current (μA)	Anodic Peak Potential (mV)	Cathodic Peak Current (μA)	Cathodic Peak Potential (mV)
CPE	3.4	380	−1.6	160
CoMOF/CPE	4.3	380	−1.9	160
1-M,3-BB/CPE	5.3	370	−1.7	235
GO/CPE	6.7	205	−4.2	0
CoMOF-GO/CPE	9.3	170	−4.2	0
CoMOF-GO/1-M,3-BB/CPE	14	130	−4.9	0

**Table 2 micromachines-13-01834-t002:** A comparison between the analytical performances of the proposed electrode with other DA sensors.

Modifier	Linear Range (µM)	LOD (nM)	References
AuNBP/MWCNTs	0.05–2700	15	[10]
Graphene	4–100	2640	[53]
Graphene-ZIF-8	3–1000	1000	[54]
Cationic surfactant Cetyltrimethylammonium Bromide	1–70	200	[55]
Ag/GO/ITO	0.1–100	200	[56]
NiO-CuO/GR	0.5–20	167	[57]
CuTRZMoO4@PPy-n	1–100	80	[58]
CuO/g-C_3_N_4_	0.2–16	60	[59]
CoMOF-GO/1-M,3-BB	0.01–300	40	This work

**Table 3 micromachines-13-01834-t003:** Voltametric sensing of DA and UA in real specimens using CoMOF-GO/1-M,3-BB/CPE. All concentrations are in µA (n = 5).

Sample	Spiked		Found		Recovery (%)		R.S.D. (%)	
	DA	UA	DA	UA	DA	UA	DA	UA
**DA injection**	0	0	4.0	-	-	-	2.9	-
	1.0	5.0	4.9	5.1	98.0	102.0	3.1	1.7
	2.0	7.5	6.1	7.3	101.7	97.3	1.9	2.3
	3.0	10.0	6.9	10.1	98.6	101.0	2.4	2.2
	4.0	12.5	8.2	12.4	102.5	99.2	2.7	3.5
**Urine**	0	0	-	3.0	-	-	-	-
	5.0	1.0	5.1	3.9	102.0	97.5	1.8	3.0
	7.0	3.0	6.8	6.1	97.1	101.7	2.6	2.0
	9.0	5.0	8.8	7.9	97.8	98.7	3.2	2.7
	11.0	7.0	11.1	10.4	100.9	104.0	2.6	2.9

## Data Availability

All the data are presented in the manuscript.

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
