# Peer review of "Simultaneous Determination of Dopamine and Uric Acid in Real Samples Using a Voltammetric Nanosensor Based on Co-MOF, Graphene Oxide, and 1-Methyl-3-butylimidazolium Bromide"

_micromachines, 2022, doi:10.3390/mi13111834_

Round 1

Reviewer 1 Report

Figure 6: Authors stated that two oxidation and reduction peaks are present at the bare CPE, however, there is only one redox pair visible at both electrodes. Baseline measurements without the presence of dopamine should be added in Figure 6 for comparison.

In Figure 8 the Tafel plot is only in the inset. Proper figure caption must be provided to void the confusion.

In Chap. 3.4, the expression “t-12” is wrong, it should be “t-1/2”. Why authors stated the diffusion coefficient for “sunset yellow”?

In Chap. 3.5, the authors wrote the detection limit of 40 nM in the text, however, the LOD of 7.1 nM is written in Table 1. Authors must explain this notable difference and provide a correction.

Authors must provide calibration SWV measurements for simultaneous detection of both dopamine and uric acid together with the resulting calibration dependencies and linear equations as a new figure. If the number of figures is limited, Figures 8 and 9 can be safely omitted as they are only of minor importance regarding the topic of the work.

Chap. 3.6: Peaks of dopamine and uric acid are not overlapping in case of low concentration, however, there is a significant overlap in higher concentration. The corresponding statement in the text must be corrected to reflect this behavior.

Figure 10 is wrongly described as Figure 9.

English language must be carefully proofread again to avoid confusing sentences, which are very often present in the manuscript.

Major revision is required for the manuscript.

Author Response

Comments and Suggestions for Authors

Figure 6: Authors stated that two oxidation and reduction peaks are present at the bare CPE, however, there is only one redox pair visible at both electrodes.

Dear reviewer, we corrected it.

Baseline measurements without the presence of dopamine should be added in Figure 6 for comparison.

Dear reviewer, we added it to the manuscript.

In Figure 8 the Tafel plot is only in the inset. Proper figure caption must be provided to void the confusion.

Dear reviewer, we corrected the figure 8 caption.

In Chap. 3.4, the expression “t-12” is wrong, it should be “t-1/2”. Why authors stated the diffusion coefficient for “sunset yellow”?

Dear reviewer, we corrected it.

In Chap. 3.5, the authors wrote the detection limit of 40 nM in the text, however, the LOD of 7.1 nM is written in Table 1. Authors must explain this notable difference and provide a correction.

Dear reviewer, we corrected it.

Authors must provide calibration SWV measurements for simultaneous detection of both dopamine and uric acid together with the resulting calibration dependencies and linear equations as a new figure. If the number of figures is limited, Figures 8 and 9 can be safely omitted as they are only of minor importance regarding the topic of the work.

Dear reviewer, Fig. 10 is the exact Figure that you proposed.

Chap. 3.6: Peaks of dopamine and uric acid are not overlapping in case of low concentration, however, there is a significant overlap in higher concentration. The corresponding statement in the text must be corrected to reflect this behavior.

Dear reviewer, it is a rule in electrochemistry that when concentration increase, the peaks will be broader, this is the reason of broad peaks is high concentrations.

Figure 10 is wrongly described as Figure 9.

Dear reviewer, we corrected it.

English language must be carefully proofread again to avoid confusing sentences, which are very often present in the manuscript.

Major revision is required for the manuscript.

Dear reviewer, we tried to improve the English language.

Reviewer 2 Report

I attach here a word document with some remarks.

Author Response

(The authors gave the same response as above.)

Round 2

Reviewer 1 Report

The authors resolved almost all issues raised by the reviewers.

However, there is one important issue related to the analysis of real samples. How is it possible that the authors did not find any uric acid in a urine sample (first row in Table 3 for a urine sample) when the concentration of uric acid in urine could be around 0.5 g/l depending on the pH of urine?? Experimental details about analyzed real samples are missing in the article. The authors must provide a new figure documenting the analysis of a real sample of urine with voltammograms and the corresponding evaluation using the standard addition method. If human urine was analyzed, the corresponding ethical statement must be added to the manuscript.

Table 1 is not formatted according to the template of the journal and the table caption is missing.

Authors sometimes use “Co-“, but with no connection to cobalt. Careful proofreading of the article is needed in some places.

Section Acknowledgements contains text from the template of the journal.

The manuscript still requires a minor revision.

Author Response

The authors resolved almost all issues raised by the reviewers.

However, there is one important issue related to the analysis of real samples. How is it possible that the authors did not find any uric acid in a urine sample (first row in Table 3 for a urine sample) when the concentration of uric acid in urine could be around 0.5 g/l depending on the pH of urine?? Experimental details about analyzed real samples are missing in the article. The authors must provide a new figure documenting the analysis of a real sample of urine with voltammograms and the corresponding evaluation using the standard addition method. If human urine was analyzed, the corresponding ethical statement must be added to the manuscript.

Dear reviewer, we are sorry or this mistake. We corrected the taable.

But it is not possible for us to get ethical statement now.

Table 1 is not formatted according to the template of the journal and the table caption is missing.

Dear reviewer, we corrected it.

Authors sometimes use “Co-“, but with no connection to cobalt. Careful proofreading of the article is needed in some places.

Dear reviewer, we used Co for cobalt.

Section Acknowledgements contains text from the template of the journal.

Dear reviewer, we deleted it.

The manuscript still requires a minor revision.

Submission Date

13 August 2022

Date of this review

15 Sep 2022 16:06:25